# ATP-Dependent Mismatch Recognition in DNA Replication Mismatch Repair

**Nianqin Zhang** [1,2] **and Yongjun Zhang** [3,*]

1   Department of Cardiology, Fuwai Hospital, National Center for Cardiovascular Diseases, Chinese Academy of Medical Sciences and Peking Union Medical College, Beijing 100037, China
2   The Second School of Clinical Medicine, Zhujiang Hospital, Southern Medical University, Guangzhou 510282, China
3   Science College, Liaoning Technical University, Fuxin 123000, China
*   Correspondence: yong.j.zhang@gmail.com

**Abstract:** Mismatch repair is a critical step in DNA replication that occurs after base selection and proofreading, significantly increasing fidelity. However, the mechanism of mismatch recognition has not been established for any repair enzyme. Speculations in this area mainly focus on exploiting thermodynamic equilibrium and free energy. Nevertheless, non-equilibrium processes may play a more significant role in enhancing mismatch recognition accuracy by utilizing adenosine triphosphate (ATP). This study aimed to investigate this possibility. Considering our limited knowledge of actual mismatch repair enzymes, we proposed a hypothetical enzyme that operates as a quantum system with three discrete energy levels. When the enzyme is raised to its highest energy level, a quantum transition occurs, leading to one of two low-energy levels representing potential recognition outcomes: a correct match or a mismatch. The probabilities of the two outcomes are exponentially different, determined by the energy gap between the two low energy levels. By flipping the energy gap, discrimination between mismatches and correct matches can be achieved. Within a framework that combines quantum mechanics with thermodynamics, we established a relationship between energy cost and the recognition error.

**Keywords:** DNA replication; mismatch repair; base pair recognition; quantum mechanics; Maxwell's demon





## 1. Introduction

DNA polymerases efficiently replicate the genome by pairing nucleotide bases with their complementary template bases, enabling the accurate transfer of genetic information during cell division. However, despite the polymerase's proofreading ability, occasional misincorporation of bases occurs, resulting in mismatches such as non-Watson/Crick base pairs and insertion/deletion errors, with an error frequency of approximately $10^{-7}$ [1]. DNA mismatch repair (MMR) [2,3] corrects these mismatches, increasing the fidelity of DNA replication by up to 1000-fold [4]. This leads to a fidelity as high as one error per $10^{10}$ base pairs [5,6].

The key enzymes involved in the canonical MMR pathway are MutS [7], MutL [8], and their homologs. MutS is an asymmetric dimer with a disc-shaped structure, featuring two channels separated by Domains I. The lower channel accommodates the DNA, while the ATPase sites are located at the top of the upper channel [9]. MutL, also a dimer, possesses N-terminal ATPase domains that can be loaded onto DNA by MutS [10]. The prevailing model of MMR suggests that MutS scans DNA for mismatches without requiring energy and, upon detection, recruits MutL, which activates the repair process [11,12]. However, this model is unable to account for all experimental observations [13]. A recent study [14] has indicated that the recognition of mismatches requires adenosine triphosphate (ATP)

and involves both MutS and MutL. In addition, certain archaeal species are associated with alternative pathways such as NucS/EndoMS [15–18] that are even less understood.

Interestingly, various MMR pathways exhibit a similar enhancement in DNA replication fidelity [19,20]. This observation implies that, despite the presence of multiple strategies to construct a recognition enzyme, they all operate within a similar threshold of recognition accuracy. In other words, the accuracy of recognition is not solely determined by the specific intricacies of the recognition process. It is plausible that a general principle, possibly in combination with energy expenditure, governs the limitations on recognition accuracy.

Traditionally, research on the mechanism of mismatch recognition has predominantly focused on the intricate mechanical aspects, such as how enzymes physically interrogate a mismatch [21,22] and the conformational changes of enzymes such as MutS/MutL. Despite numerous speculations regarding the mechanism of mismatch recognition [23], there remains a significant research gap in understanding the recognition accuracy. This knowledge gap is due to both our limited understanding of MMR [24] and the mysterious nature of molecular-scale recognition processes within the realm of physics [25,26]. The underlying mystery may be linked to the fundamental principles of quantum physics [27,28].

In this study, we explored the potential connection between mismatch recognition and quantum mechanics. We also distinguished between passive recognition and active recognition, highlighting the possible role of ATP utilization. Another critical aspect affecting replication fidelity is the risk of mistakenly identifying a correct match as a mismatch. Surprisingly, this aspect has received limited attention in the existing literature. We proposed an approach to investigating this factor and integrating it into the overall fidelity, marking the first attempt to do so.

## 2. Passive Recognition versus Active Recognition

The mystery surrounding recognition and measurement can be traced back to 1867, when Maxwell introduced a thought experiment called Maxwell's demon [29]. In this experiment, a hypothetical demon appears to challenge the second law of thermodynamics by performing cyclic measurements without expending any energy. To reconcile this paradox and preserve the second law of thermodynamics, researchers established two rules: (1) information can be converted into free energy [30]; (2) erasing information requires energy [31,32]. These rules make the demon unable to sustain its operations without consuming energy.

It is important to note that energy cost can also occur during the process of measurement [33]. However, the mechanisms underlying molecular-level measurements remain mysterious. One approach to advancing research in this field is to investigate specific examples found in biology. Mismatch recognition in DNA mismatch repair (MMR) is one such example because recognitions, in essence, involve the same principles as measurements or information acquisition [34]. MMR is particularly well-suited for such studies due to the following reasons: (1) the mismatch recognition process in MMR is a standalone process and can be studied separately from other processes; (2) a considerable amount of knowledge has been accumulated regarding MMR; (3) data on DNA replication fidelity are available so that a proposed model can be validated.

Recognition processes involve energy and are partly governed by the principles of thermodynamics. Two categories of recognition can be identified: passive recognition and active recognition. Passive recognition operates within an equilibrium framework, where a mismatch and a correct match elicit different interactions with the enzyme responsible for recognition, resulting in distinct affinities and free energies. In an equilibrium system, states with lower free energy are more probable, following the Boltzmann distribution. It is possible to optimize the enzyme's structure to achieve low free energy when bound to a mismatch and high free energy when bound to a correct match. However, passive recognition has inherent limitations: (1) The enzyme's movement is random and lacks directionality, undergoing Brownian motion in equilibrium [35]; (2) the accuracy of recognition is limited due to the limited availability of free energy.

Active recognition relies on non-equilibrium properties sustained through the consumption of energy, with ATP serving as the energy source in this study. Active recognition offers several advantages: The enzyme can exhibit directional motion, scanning each base pair in a one-way fashion, avoiding time-consuming back-and-forth movements. The utilization of ATP enables high recognition accuracy, as ATP can provide energy higher than the free energy associated with equilibrium. ATP enables auxiliary operations that would otherwise be impossible, such as inducing conformational changes in double-stranded DNA through bending [36,37].

Passive recognition processes may not incur an immediate energetic cost during the initial recognition event. However, the energy cost is deferred and occurs later during the preparation for subsequent recognition events, such as the restoration of enzyme conformation. Therefore, passive recognitions are not inherently more energetically efficient than active recognitions, particularly in cyclic processes. While passive recognition may play a role in certain biological activities, such as antibody-antigen recognition, active recognition is necessary in DNA replication, where both speed and fidelity are crucial factors.

### 3. Active Recognition Framework

Since the mechanisms and enzymes involved in MMR are still being studied and there is no specific enzyme with well-established characteristics, our study focused on a hypothetical enzyme that we refer to as Enz. Enz shares some features with MutS/MutL enzymes but is specialized solely in mismatch recognition. Furthermore, we considered a simplified scenario in which the DNA strand consists of only two types of base pairs: the A–T correct match, which represents all correct matches, and the G–T mismatch [38,39], which represents all types of mismatches.

Enz is a molecular machine that operates through the utilization of ATP, similar to other protein motors and molecular machines found in nature. In a sense, these proteins can be likened to Maxwell's demon [40]. The design and optimization of Enz can be achieved through evolutionary processes, ensuring its effectiveness in increasing replication fidelity by 1000-fold. However, it is important to note that the ultimate goal of improving replication fidelity must still adhere to the fundamental laws of physics. Therefore, instead of studying the specific structure of Enz, our focus was on understanding its functions and how they are supported by the principles and laws of physics.

Enz utilizes two ATP molecules for a single recognition event. Figure 1 illustrates the framework depicting the functioning of Enz. The framework involves a complex comprising Enz, a base pair, and ATP/ADP molecules. This complex undergoes a sequence of states labeled as $S$, $B$, $W$, $R$, and so forth. Each state is characterized by a conformation and a distinct energy. When a complex is formed, it inherits the conformation of Enz but changes the corresponding energy under the influence of the interaction between Enz and the base pair.

The complex depicted in Figure 1a contains an A–T correct match. Initially, it is in state $S$, characterized by a low energy level and an open conformation. Upon binding to two ATP molecules, the complex transitions to state $B$, where one ATP is bound and the other undergoes hydrolysis, resulting in a closed conformation. This transition marks the initiation of the recognition process, as the complex reaches its highest energy level. At the moment of hydrolysis of the remaining ATP, recognition takes place, leading to one of two states: state $R$ or state $W$. State $R$ corresponds to the recognition of the A–T as a correct match, while state $W$ corresponds to the recognition of the A–T as a mismatch. These states exhibit distinct conformations, which subsequently dictate the following processes. In state $R$, Enz undergoes a directional translocation towards the next base pair, initiating a new round of recognition. In contrast, in state $W$, Enz either signals for repair or remains in state $W$, which itself serves as the signal.

Similarly, the recognition process for the G–T mismatch follows an analogous series of states: $S'$, $B'$, $W'$, and $R'$. While the states depicted in Figure 1a,b are related, they may not have the same energy levels due to the involvement of different base pairs. For

instance, state $R$ and state $R'$ share the same conformation and subsequent processes, but their energies differ.

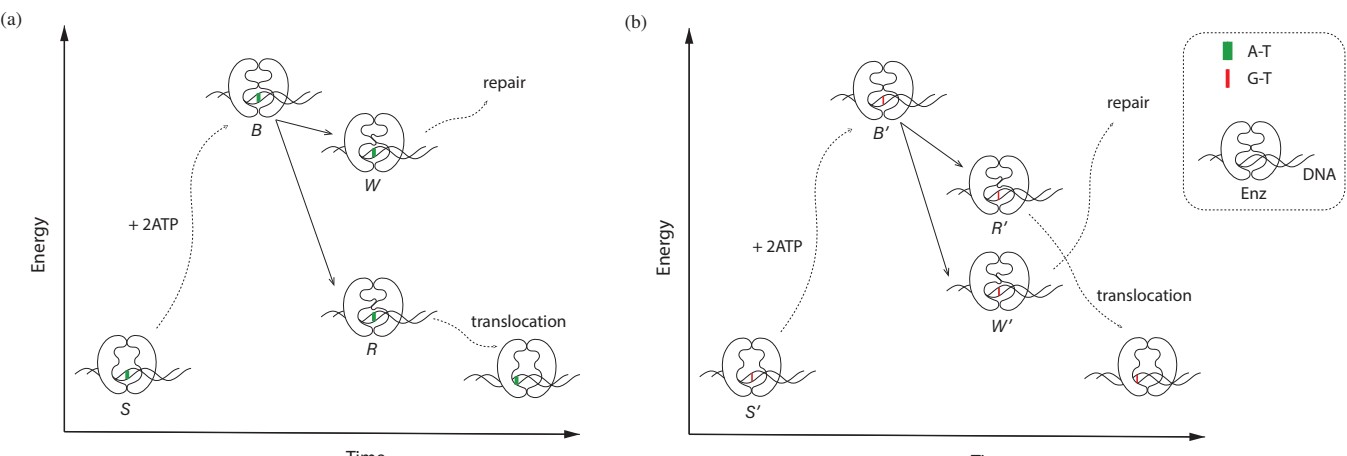

**Figure 1.** A theoretical framework of MMR, focusing on a hypothetical enzyme we refer to as Enz, which shares basic characteristics with MutS/MutL. Enz undergoes a series of conformational changes driven by the energy derived from ATP. The subject of the framework is a complex composed of Enz, a base pair, and ATP/ADP molecules. The total energy of the complex, represented on the vertical axis, includes the chemical energy stored in ATP. Dotted lines indicate multi-step changes or blurry details between states, while solid lines represent quantum transitions that occur during base pair recognition. Enz only takes certain configurations, which are eigenstate solutions to the Schrödinger equation; therefore, a change in configuration is a quantum transition. The coupling between Enz and the base pair is a quantum coupling, which significantly influences the energy level of the complex. (**a**) A–T correct match. In state $S$, the complex takes the lowest energy level and an open conformation. Upon binding to two ATP molecules, hydrolysis of one ATP leads to a closed conformation, resulting in state $B$. At this stage, the complex reaches its highest energy level, primed for the recognition process. The transition occurs upon hydrolysis of the second ATP, leading to either state $W$ or state $R$. In state $R$, Enz slides to the next base pair, initiating a new round of recognition. In state $W$, Enz signals for repair. (**b**) G–T mismatch. The complex undergoes a similar dynamic process, although the energy levels are rearranged due to the involvement of a different base pair.

## 4. Quantum Mechanics

Let us further investigate the first scenario presented in Figure 1a, where the base pair is A–T. Our focus will be on the recognition transition, which serves as the core of the recognition process and involves bifurcation, resulting in two distinct recognition outcomes. Given its size, the complex can be regarded as a quantum system. States $B$, $R$, and $W$ are quantum states, each associated with a discrete energy level: $E_B$, $E_R$, and $E_W$, respectively. The recognition process is characterized by a quantum transition (Figure 2a), accompanied by a sudden change in the conformation of the complex. ATP hydrolysis itself is a quantum process. When Enz combines with ATP, they form a quantum system via quantum coupling. As a result, the process of ATP hydrolysis becomes a quantum process involving the entire complex.

The complex is not only a quantum system but is also a thermodynamic system. It is composed of thousands of atoms, and each atom undergoes random thermal vibrations, characterized by thermal energy on the order of $kT$, where $k$ is the Boltzmann constant and $T$ is the absolute temperature of the surrounding water. Even when the complex is in state $B$, the specific thermal vibrations of its atoms can assume various configurations, resulting in multiple ($N_B$) quantum states denoted as $B_i$ where $i = 1, 2, \cdots$. In accordance with thermodynamics, each quantum state $B_i$ corresponds to a microscopic state, and all microscopic states are equally probable. Consequently, we calculate the average over all possible initial states $B_i$ using $\frac{1}{N_B} \sum$. Similarly, state $R$ corresponds to multiple ($N_R$)

quantum states $R_j$. Each $R_j$ is a distinct quantum state and could potentially be the actual outcome of the transition. Therefore, we sum over all possible final states $R_j$. Therefore, for the transition from state $B$ to state $R$, the transition probability can be expressed as

$$P_R \propto \frac{1}{N_B} \sum_{i=1}^{N_B} \sum_{j=1}^{N_R} |\langle R_j|H|B_i\rangle|^2. \tag{1}$$

Here, $P_R$ represents the transition probability and $\langle R_j|H|B_i\rangle$ denotes the matrix element of the Hamiltonian $H$, which characterizes the dynamics of the complex [41]. If the term $|\langle R_j|H|B_i\rangle|^2$ is independent of $i$ and $j$, the equation can be simplified to

$$P_R \propto |\langle R|H|B\rangle|^2 N_R. \tag{2}$$

If $|\langle R_j|H|B_i\rangle|^2$ is not independent of $i$ and $j$, we can still arrive at the same equation by taking $|\langle R|H|B\rangle|^2$ as the average of $|\langle R_j|H|B_i\rangle|^2$.

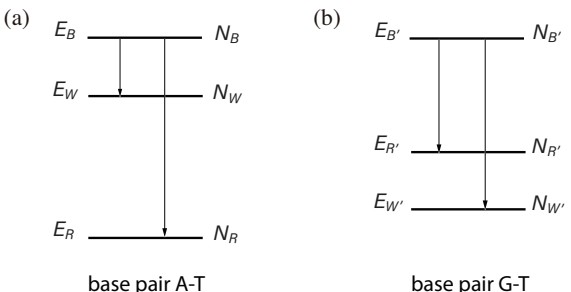

(a) base pair A-T  (b) base pair G-T

**Figure 2.** Base pair recognition through quantum transitions. A quantum transition involves a jump from the highest energy level to a lower energy level. However, quantum transitions between low energy levels are suppressed. (**a**) In the case of an A–T correct match, the number of microscopic states associated with state $R$ (denoted as $N_R$) is greater than the number of microscopic states associated with state $W$ (denoted as $N_W$), meaning that $N_R > N_W$. When the system is in state $R$, it actually exists in a specific microscopic state, $R_j$, where $j = 1, 2, \cdots$, at any given time. $R_j$ specifies the thermal vibrations in the system's atoms. We can regard $R_j$ as a distinct quantum state and $N_R$ as a degeneracy. Similarly, we introduce quantum states $B_i$ and $W_k$. Before the transition occurs, the system is in a specific quantum state, such as $B_1$. According to quantum mechanics, when the transition takes place, all possible transition channels occur simultaneously. This implies that the result of the transition could be any of $R_1, R_2, \cdots, R_{N_R}, W_1, W_2, \cdots,$ or $W_{N_W}$, each with similar probabilities given that the Hamiltonian does not strongly differentiate between different transition channels. Therefore, if $N_R > N_W$, the transition result is more likely to be state $R$ than state $W$. The value of $N_R$ can be determined from the entropy of state $R$, which is related to the thermal energy of state $R$. For the transition $B \to R$, the thermal energy of state $R$ increases by $E_B - E_R$. Thus, we have $\ln N_R \propto E_B - E_R$. Similarly, we have $\ln N_W \propto E_B - E_W$. Consequently, $N_R$ is greater than $N_W$. (**b**) In the case of the G–T mismatch, the energy level of $W'$ is lower than that of $R'$, resulting in $N_{W'} > N_{R'}$. As a result, the transition from state $B'$ is more likely to lead to state $W'$ rather than state $R'$.

The value of $N_R$ is related to energy. When state $B$ transitions to state $R$, there is a release of energy $\Delta E = E_B - E_R$. This energy is absorbed by the complex and converted into heat, leading to an increase in entropy: $\Delta S = \Delta E / T$. Consequently, we have the relationship

$$S_R - S_B = \frac{E_B - E_R}{T}, \tag{3}$$

where $S_R$ and $S_B$ represent the entropies of states $R$ and $B$, respectively. The entropy $S$ is connected to the number of microscopic states, $\Omega$, through the Boltzmann formula, $S = k \ln \Omega$. Thus, we can express $S_B$ as $S_B = k \ln N_B$ and $S_R$ as $S_R = k \ln N_R$, which leads to

$$N_R = N_B \exp\left(\frac{E_B - E_R}{kT}\right). \tag{4}$$

For the sake of simplicity, we assumed that the complex is at the same temperature as the surrounding water. However, in reality, the heat generated during the transition would slightly increase the temperature of the complex. Fortunately, the temperature increase is not significant. After the transition, the complex dissipates the excess heat to the surrounding water, returning to its original temperature and entropy state in preparation for the next recognition event, as illustrated in Figure 3.

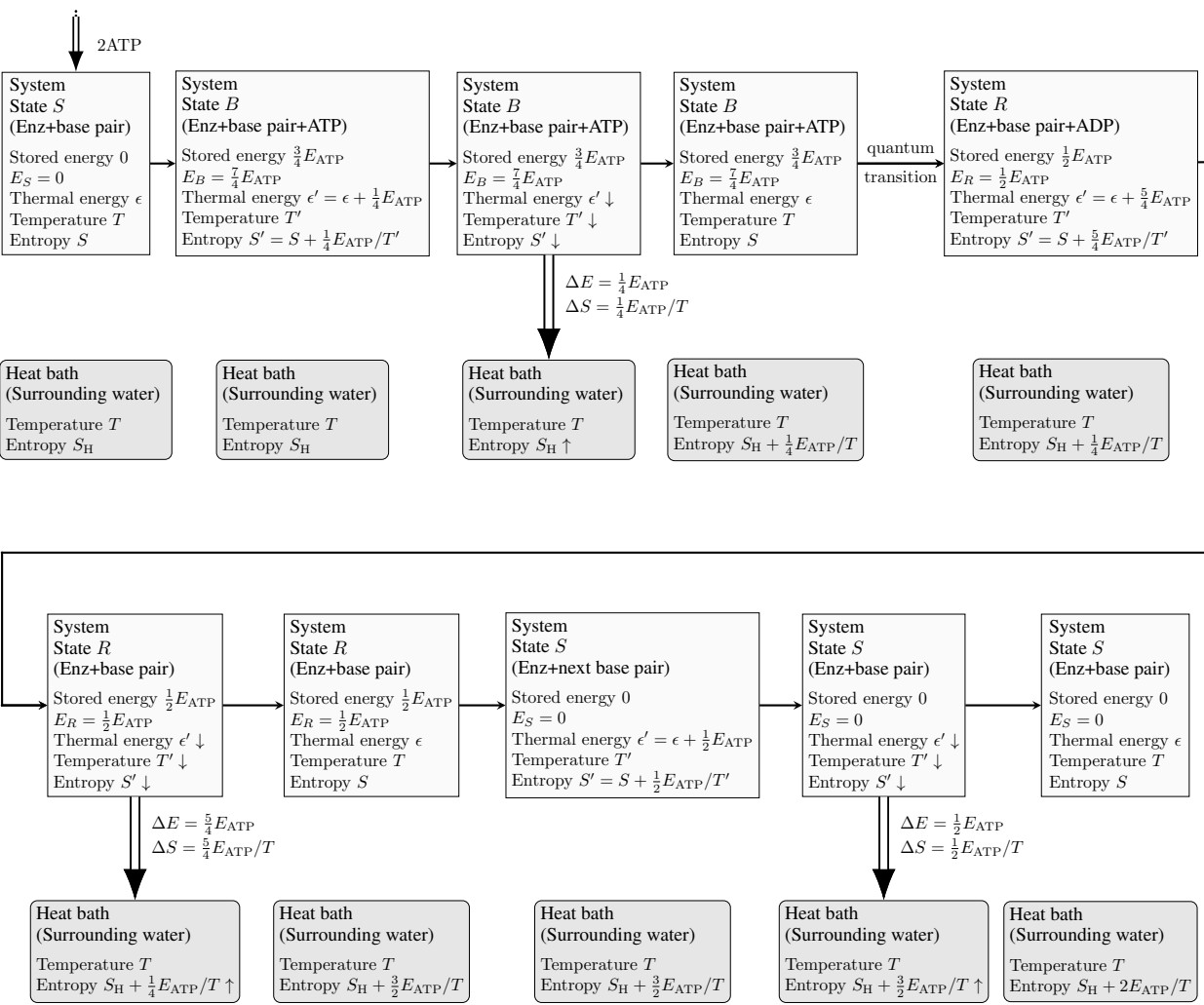

**Figure 3.** Schematic of thermodynamics associated with the quantum transition from state $B$ to state $R$. The quantities shown in the figure are purely illustrative examples. The system consists of Enz, a base pair, and ATP/ADP molecules, while the surrounding water serves as the heat bath at room temperature, $T$. In this particular example, state $B$ stores an energy of $\frac{3}{4}E_{\text{ATP}}$ plus the second ATP. Thus, $E_B = \frac{7}{4}E_{\text{ATP}}$. State $R$ holds an energy of $\frac{1}{2}E_{\text{ATP}}$, which is reserved for Enz translocation. Consequently, the energy available for the quantum transition is $\frac{5}{4}E_{\text{ATP}}$. The atoms of the system undergo random vibrations governed by the laws of thermodynamics. The energy released during the quantum transition converts into the thermal energy of the system, resulting in an increase in entropy by $\frac{5}{4}E_{\text{ATP}}/T'$, where $T'$ represents the increased temperature of the system. Assuming $E_{\text{ATP}} = 20\,kT$, we have $E_S = 0$, $E_B = 35\,kT$, and $E_R = 10\,kT$. Consequently, the thermal energy of the system increases by $25\,kT$ during the quantum transition. If Enz consists of 1000 atoms, each atom would only acquire a thermal energy of $0.025\,kT$, resulting in $T' \approx T$. Similar analyses can be conducted for the other three quantum transitions.

The complex possesses two types of energy: chemical energy associated with the energy levels depicted in Figures 1 and 2, and heat, which is linked to the thermal vibrations of the atoms within the complex. In state *B*, the chemical energy ($E_B$) is composed of two components. One component corresponds to the energy stored in the second ATP, while the other component arises from the potential energy stored in the conformation of the complex when the first ATP is hydrolyzed. Consequently, the energy of state *B* is greater than that of one ATP but less than that of two ATPs. A sufficiently high $E_B$ is necessary to achieve a 1000-fold improvement in MMR fidelity.

The reverse transition from state *R* to state *B* is suppressed. This is due to the higher temperature of the complex associated with state *R*, resulting in heat flowing from the complex to the surrounding water. This asymmetry in heat flow breaks the time-reversal symmetry and leads to the suppression of the reverse transition. Similarly, the reverse transition from state *B* to state *S* is also suppressed as a consequence of the hydrolysis of the first ATP, followed by the dissipation of the released heat. The involvement of non-equilibrium thermodynamics is necessary so that time gains direction in which MMR can proceed.

Similarly, for the transition from state *B* to state *W*, the transition probability can be expressed as

$$P_W \propto \frac{1}{N_B} \sum_{i=1}^{N_B} \sum_{j=1}^{N_W} |\langle W_j | H | B_i \rangle|^2, \tag{5}$$

where $N_W$ represents the number of microscopic states associated with state *W*. Or, we can simply have

$$P_W \propto |\langle W | H | B \rangle|^2 N_W. \tag{6}$$

The value of $N_W$ is determined by the energy difference between states *B* and *W* as

$$N_W = N_B \exp\left(\frac{E_B - E_W}{kT}\right). \tag{7}$$

It is important to note that the transition from state *W* to state *R* is also possible if the subsequent process of state *W* can wait. This would potentially improve the recognition accuracy. However, in the context of DNA mismatch repair, state *W* cannot afford to wait because both recognition speed and accuracy are crucial. The subsequent process needs to be triggered promptly. As a result, this transition is effectively suppressed.

Since the base pair to be recognized here is the correct match A–T, we expect the recognition to result in state *R* rather than state *W*. To quantify this, we introduced a small parameter,

$$\xi^{AT} = \frac{P_W}{P_R}. \tag{8}$$

We refer to this parameter as a recognition error. It is important to note that recognition error should not be confused with error rate or error frequency. The error rate, denoted as $\eta$, is defined in this case as the probability of transitioning to state *W* relative to the total probability of transitioning to either state *W* or state *R*, i.e.,

$$\eta = \frac{P_W}{P_W + P_R}. \tag{9}$$

Given a typical small recognition error $\xi$, the corresponding error rate $\eta$ is also small and $\eta \approx \xi$. However, there are cases where these two quantities can significantly differ. An extreme example is when Enz is dysfunctional, resulting in a completely random recognition process. In this case, we would have $\xi = 1$ ($\xi = 100\%$) and $\eta = 0.5$ ($\eta = 50\%$).

The two transitions, $B \rightarrow R$ and $B \rightarrow W$, represent different outcomes of the same quantum process, which involves the evolution of the initial quantum state $B_i$ over time. The evolution of a quantum state is described by the Schrödinger equation,

$$i\hbar \frac{d}{dt}|\psi\rangle = H|\psi\rangle, \tag{10}$$

where $\hbar$ is the reduced Planck constant, and $|\psi\rangle$ represents the quantum state. The solution to the Schrödinger equation, as provided in [42], is

$$|\psi(t)\rangle = e^{-\frac{i}{\hbar}Ht}|\psi(0)\rangle, \tag{11}$$

where $e^{-\frac{i}{\hbar}Ht}$ is the time evolution operator. This equation describes the time evolution of the initial quantum state $B_i$, represented as $|B_i\rangle$ in Dirac notation, as

$$|B_i\rangle \xrightarrow[\text{evolution}]{t} e^{-\frac{i}{\hbar}Ht}|B_i\rangle. \tag{12}$$

For a short time interval $t$, we can make the following approximation,

$$\begin{aligned}|B_i\rangle &\xrightarrow[\text{evolution}]{t} e^{-\frac{i}{\hbar}Ht}|B_i\rangle \approx \left(1 - \frac{i}{\hbar}Ht\right)|B_i\rangle \\ &= |B_i\rangle - \frac{i}{\hbar}t \sum_{j=1}^{N_B}|B_j\rangle\langle B_j|H|B_i\rangle - \frac{i}{\hbar}t\sum_{j=1}^{N_R}|R_j\rangle\langle R_j|H|B_i\rangle - \frac{i}{\hbar}t\sum_{j=1}^{N_W}|W_j\rangle\langle W_j|H|B_i\rangle.\end{aligned} \tag{13}$$

The smallness of $t$ arises from the frequent interruptions of the unitary time evolution of the quantum wave function by thermal vibrations which frequently cause the wave function to collapse. A collapse can result in any of $B_i$, $B_j$, $R_j$, or $W_j$. If the outcome is $B_i$ or $B_j$, the transition restarts and $t$ is reset to 0. If the outcome is $R_j$ or $W_j$, the transition finishes. Equation (13) reveals that Equations (1) and (5) share a common factor, which cancels in Equation (8), yielding the expression

$$\xi^{\text{AT}} = \frac{|\langle W|H|B\rangle|^2}{|\langle R|H|B\rangle|^2} \exp\left\{-\frac{E_W - E_R}{kT}\right\}. \tag{14}$$

For the second scenario, where the base pair is G–T (Figure 1b), the transition occurs from state $B'$ to either state $W'$ or state $R'$ (Figure 2b). The probabilities of these transitions are denoted as $P_{W'}$ and $P_{R'}$, respectively. In this case, the recognition result is expected to be $W'$ rather than $R'$. To quantify the recognition error for G–T, we introduced a second small parameter,

$$\xi^{\text{GT}} = \frac{P_{R'}}{P_{W'}}, \tag{15}$$

which can be calculated as

$$\xi^{\text{GT}} = \frac{|\langle R'|H|B'\rangle|^2}{|\langle W'|H|B'\rangle|^2} \exp\left\{-\frac{E_{R'} - E_{W'}}{kT}\right\}, \tag{16}$$

where $E_{R'}$ and $E_{W'}$ denote the energies associated with states $R'$ and $W'$, respectively, and $\langle R'|H|B'\rangle$ and $\langle W'|H|B'\rangle$ are transition matrix elements.

We observe that the Hamiltonian matrix elements have a similar magnitude for the transitions $W \rightarrow B$ and $W' \rightarrow B'$, as well as for the transitions $R \rightarrow B$ and $R' \rightarrow B'$. This is because: (i) The major part of the complex is Enz in both scenarios. (ii) State $B$ and state $B'$ have identical Enz conformation. (iii) State $W$ and state $W'$ also have identical Enz conformation. Based on these observations, we can have

$$|\langle W|H|B\rangle|^2 \approx |\langle W'|H|B'\rangle|^2 \tag{17}$$

and

$$|\langle R|H|B\rangle|^2 \approx |\langle R'|H|B'\rangle|^2. \tag{18}$$

Therefore, we have

$$\frac{|\langle W|H|B\rangle|^2}{|\langle R|H|B\rangle|^2} \times \frac{|\langle R'|H|B'\rangle|^2}{|\langle W'|H|B'\rangle|^2} \approx 1. \tag{19}$$

As we shall see in the next section, what matters is the product of $\xi^{AT}$ and $\xi^{GT}$, rather than their individual values. The coefficients $\frac{|\langle W|H|B\rangle|^2}{|\langle R|H|B\rangle|^2}$ and $\frac{|\langle R'|H|B'\rangle|^2}{|\langle W'|H|B'\rangle|^2}$ will cancel each other out in the calculation of the fidelity of DNA replication. Additionally, the two coefficients themselves may both be approximately 1. Since they cannot be small simultaneously as desired, it is reasonable to assume them to be approximately equal to each other and, consequently, approximately equal to 1. Therefore, we can simplify Equations (14) and (16) by dropping these coefficients and write

$$\xi^{AT} = \exp\left\{-\frac{E_W - E_R}{kT}\right\}, \tag{20}$$

and

$$\xi^{GT} = \exp\left\{-\frac{E_{R'} - E_{W'}}{kT}\right\}. \tag{21}$$

To provide a concrete example and illustrate the underlying physics, let us consider the following scenario at room temperature (Figure 4b),

$$\begin{cases} E_W - E_R = 18\,kT, \\ E_{R'} - E_{W'} = 5\,kT. \end{cases} \tag{22}$$

Using Equations (20) and (21), we can evaluate the values of $\xi^{AT}$ and $\xi^{GT}$ as

$$\xi^{AT} \sim 10^{-8}, \ \xi^{GT} \sim 10^{-2}. \tag{23}$$

Thus, the recognition errors for the A–T and G–T base pairs are obtained.

The energy provided by an ATP molecule for the recognition process may not be exactly equal to the Gibbs free energy of ATP, but their values are expected to be close. The Gibbs free energy per ATP molecule is approximately 20 $kT$ (around 50 kJ/mol) at room temperature [43]. Therefore, each ATP molecule should be able to provide approximately 20 $kT$ of energy to Enz.

In the asynchronous hydrolysis of the two ATP molecules (Figure 3), the first ATP hydrolysis event triggers a conformational change in Enz to prepare it for the recognition process. The energy provided by the first ATP may be more than what is required for the conformational change. The excess energy could be stored in the conformation $B/B'$ to be utilized later in conjunction with the energy from the hydrolysis of the second ATP. As a result, the energy values $E_B$ and $E_{B'}$ could be higher than the energy of a single ATP molecule. These values are expected to fall between 20 $kT$ and 40 $kT$. Additionally, it is convenient to assume that $E_B \approx E_{B'}$. This assumption could be realistic because it is desirable for $E_B$ and $E_{B'}$ to be as high as possible, which, in turn, ensures that they are approximately equal.

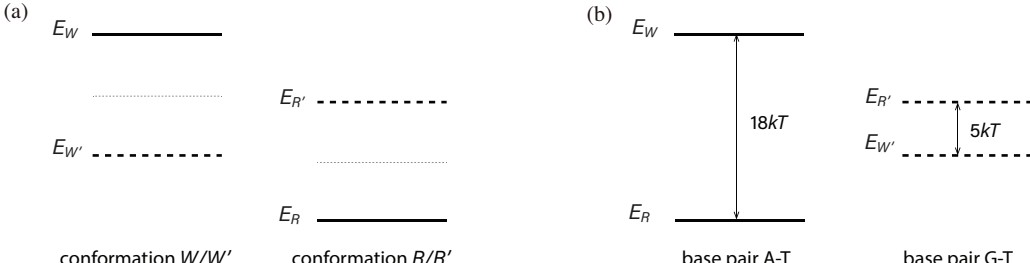

**Figure 4.** An example of energy-level shifts and energy gaps. (**a**) Energy-level shifts. The quantum interactions between Enz and base pairs induce shifts in the energy levels. When Enz takes the conformation $W(W')$ and combines with no base pairs, its energy level takes a certain value as indicated by the dotted line. However, when Enz combines with a base pair, the energy level undergoes a shift. Specifically, the A–T base pair causes the energy level to shift upward, resulting in a high value of $E_W$, while the G–T base pair causes it to shift downward, resulting in a low value of $E_{W'}$. Similar discussions apply to the conformation $R(R')$. However, for this case, the A–T base pair shifts the energy level downward, resulting in a low value of $E_R$, while the G–T base pair shifts it upward, resulting in a high value of $E_{R'}$. These energy-level shifts are implemented in the structure of Enz through the processes of evolution. (**b**) Two energy gaps generated from the energy-level shifts. We have two energy gaps here, associated with the A–T base pair and the G–T base pair, respectively. These energy gaps determine the sensitivity and discrimination ability of Enz. In this example, at room temperature, we have $E_W - E_R = 18\,kT$ and $E_{R'} - E_{W'} = 5\,kT$. The corresponding recognition errors are estimated to be $\xi^{\text{AT}} \sim 10^{-8}$ and $\xi^{\text{GT}} \sim 10^{-2}$ for the A–T and G–T base pairs, respectively.

## 5. Reshuffle Energy Levels

Enz possesses the ability to discriminate between correct base pair matches and mismatches, and this ability is rooted in quantum mechanics. To achieve this discrimination, it is necessary to manipulate the energy gap between conformations $R/R'$ and $W/W'$ to result in $E_R < E_W$ and $E_{R'} > E_{W'}$.

Enz interacts differently with various base pairs, and it uses these variations in its recognition mechanism to induce distinct energy-level shifts, as depicted in Figure 4a. For example, Enz in the $W/W'$ conformation exhibits specific interactions with A–T and G–T, causing an upward shift of the energy level $E_W$ and a downward shift of $E_{W'}$. Consequently, there is an energy-level shift of $E_W - E_{W'}$. Similarly, an energy-level shift of $E_R - E_{R'}$ is created, but in the opposite direction. The magnitudes of these energy-level shifts play a crucial role in determining the sensitivity of Enz. Ideally, these shifts should be maximized, but there are inherent limits to their values, which should have been reached through the process of evolution.

The interactions between Enz and base pairs are quantum interactions, which have the inherent ability to induce energy-level shifts [44–47]. Without quantum interactions, there would be no energy-level shifts, and we would be stuck with $E_W = E_{W'}$ and $E_R = E_{R'}$. Quantum interactions can either raise or lower energy levels, depending on the specific conformation and the specific base pair. The strength of the quantum interaction determines the magnitude of the energy-level shift. One approach to enhancing the strength of the quantum interaction is to optimize the conformation of Enz, enabling it to have increased contact with the base pair. For instance, motifs can be inserted into the grooves of double-stranded DNA to facilitate stronger interactions between Enz and the base pair.

Since the energy-level shifts $E_W - E_{W'}$ and $E_{R'} - E_R$ have reached their limits, they can be considered as constants. Therefore, we have the relationship,

$$(E_W - E_{W'}) + (E_{R'} - E_R) = \text{constant}. \tag{24}$$

When this constant is divided into two energy gaps, $E_W - E_R$ and $E_{R'} - E_{W'}$, they will mutually constrain each other. Consequently, the values of $\xi^{\text{AT}}$ and $\xi^{\text{GT}}$ are also interdependent. If one value becomes smaller, the other unavoidably becomes larger.

For instance, if we change the splitting in Equation (22) to

$$\begin{cases} E_W - E_R = 16\,kT, \\ E_{R'} - E_{W'} = 7\,kT, \end{cases} \tag{25}$$

as shown in Figure 5, the values in Equation (23) will change to

$$\xi^{AT} \sim 10^{-7}, \ \xi^{GT} \sim 10^{-3}. \tag{26}$$

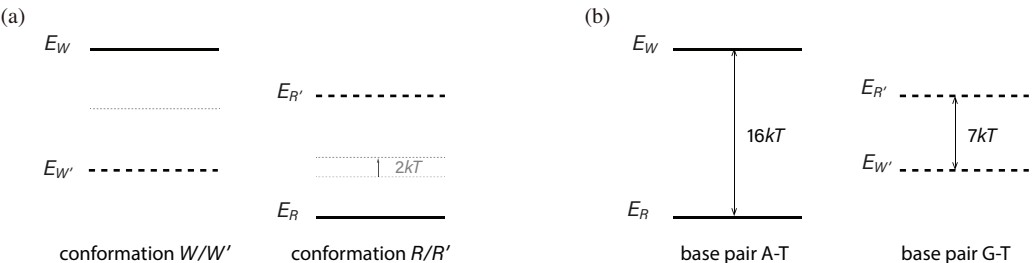

**Figure 5.** Another example of energy-level shifts and energy gaps. (**a**) Two energy-level shifts. The magnitudes of the energy-level shifts are the same as those in the previous example, representing the limits of Enz. However, in this case, both $E_{R'}$ and $E_R$ have been shifted up by $2\,kT$, which can be easily achieved by introducing an additional $2\,kT$ of chemical energy into the Enz conformation $R/R'$, without involving the interaction between Enz and base pairs. (**b**) Two generated energy gaps. The energy gaps have changed accordingly. The new energy gaps are $E_W - E_R = 16\,kT$ and $E_{R'} - E_{W'} = 7\,kT$, resulting in recognition errors of $\xi^{AT} \sim 10^{-7}$ and $\xi^{GT} \sim 10^{-3}$, respectively.

Enz's recognition ability can be quantified too. For this, we introduced a parameter,

$$\Xi = \xi^{AT} \xi^{GT}. \tag{27}$$

When $\Xi = 0$, it implies a perfect Enz. Conversely, when $\Xi = 1$, it indicates that Enz is completely dysfunctional. This can occur in the following scenarios:

- Recognition result is completely random, i.e., $\xi^{AT} = 1$ and $\xi^{GT} = 1$.
- Recognition always results in conformation $R/R'$, i.e., $\xi^{AT} = 0$ and $\xi^{GT} = \infty$, or $\frac{1}{0}$.
- Recognition always results in conformation $W/W'$, i.e., $\xi^{AT} = \infty$ and $\xi^{GT} = 0$.

When the limits on Enz's energy-level shifts are given, the value of $\Xi$ is fixed. The two examples in Figures 4 and 5 share the same limits, and therefore, the same $\Xi = 10^{-10}$. However, even when Enz's energy-level shifts are fixed, Enz's energy gaps are yet to be determined and can be adjusted relative to each other. Therefore, $\xi^{AT}$ and $\xi^{GT}$ can still be optimized, as we will show in an example.

## 6. Fidelity Improvement

During the MMR process, when a match is recognized as a mismatch, it is removed from the nascent strand along with thousands of neighboring base pairs. Subsequently, resynthesis occurs, which may introduce new errors [48,49]. The error rate of resynthesis is expected to be in the same order of magnitude as the fidelity of DNA replication prior to MMR, because the two processes use similar polymerases.

Let us use Enz to demonstrate how MMR may improve the overall fidelity of DNA replication. Four key elements shall be involved, each associated with a parameter. Let us set these parameters as follows for demonstration purposes:

- The recognition ability of Enz is $\xi^{AT} \xi^{GT} = 10^{-10}$.
- The G–T error rate before MMR is $\eta = 10^{-8}$.
- Each repair process excises $10^3$ bases.
- The error rate of resynthesis is $10^{-7}$.

The first element indicates that the limits on Enz's energy-level shifts are given, as shown in Figure 4 or Figure 5. However, the values of $\xi^{AT}$ and $\xi^{GT}$ are yet to be optimized. Combining the last three elements gives rise to the G–T error rate after MMR, which can be expressed as:

$$\eta = 10^{-8}\frac{\xi^{GT}}{1+\xi^{GT}} + \left[10^{-8}\frac{1}{1+\xi^{GT}} + \left(1-10^{-8}\right)\frac{\xi^{AT}}{1+\xi^{AT}}\right] \times 10^{-7} \times 10^3. \quad (28)$$

The mismatches after MMR arise from two different sources. Some mismatches result from mistaking a mismatch as a correct match, while others are due to errors introduced during resynthesis. When Enz encounters a G–T mismatch, with a probability of $10^{-8}$, it either correctly recognizes it with a probability of $\frac{1}{1+\xi^{GT}}$ or incorrectly recognizes it with a probability of $\frac{\xi^{GT}}{1+\xi^{GT}}$. When Enz encounters an A–T match, with a probability of $1-10^{-8}$, it either correctly recognizes it with a probability of $\frac{1}{1+\xi^{AT}}$ or incorrectly recognizes it with a probability of $\frac{\xi^{AT}}{1+\xi^{AT}}$. Each correctly recognized G–T and incorrectly recognized A–T is repaired, resulting in the removal and resynthesis of $10^3$ neighboring base pairs, which may introduce $10^{-7} \times 10^3$ new errors.

Approximately, Equation (28) can be simplified to

$$\eta \approx 10^{-8}\xi^{GT} + 10^{-12} + 10^{-4}\xi^{AT}. \quad (29)$$

Considering the constraint $\xi^{AT}\xi^{GT} = 10^{-10}$, we can choose $\xi^{AT} = 10^{-7}$ and $\xi^{GT} = 10^{-3}$ to achieve the best fidelity, resulting in $\eta \sim 10^{-11}$.

However, it should be noted that Enz does not always improve fidelity in all scenarios, as it may mistake A–T base pairs as G–T base pairs. A particularly extreme example is when there are no G–T mismatches present before MMR, meaning that $\eta = 0$ instead of $\eta = 10^{-8}$. In such a case, after MMR, the error rate would become

$$\eta = \frac{\xi^{AT}}{1+\xi^{AT}} \times 10^{-7} \times 10^3. \quad (30)$$

For the previous example where $\xi^{AT} = 10^{-7}$, this leads to $\eta \sim 10^{-11}$. Thus, the fidelity worsens from $\eta = 0$ to $\eta = 10^{-11}$. This deterioration in fidelity occurs only because of the assumption that even correct matches need to be recognized in the MMR process, and the recognition error is not zero. This scenario could be an interesting subject for future experiments to validate.

Furthermore, this assumption implies that there is a limit to the improvement in fidelity when MMR is performed repeatedly. By comparing it to Equation (28), a recursion relation for the error rate after each round of MMR can be derived as

$$\begin{aligned} \eta_{n+1} &= \eta_n\frac{\xi^{GT}}{1+\xi^{GT}} + \left[\eta_n\frac{1}{1+\xi^{GT}} + (1-\eta_n)\frac{\xi^{AT}}{1+\xi^{AT}}\right] \times 10^{-7} \times 10^3 \\ &\approx (\xi^{GT} + 10^{-4})\eta_n + 10^{-4}\xi^{AT}, \end{aligned} \quad (31)$$

where $\eta_n$ and $\eta_{n+1}$ are the error rates before and after a round of MMR, respectively. The limit of the error rate after repeated MMR is

$$\eta \approx \frac{10^{-4}\xi^{AT}}{1-\xi^{GT}-10^{-4}} \approx 10^{-4}\xi^{AT}. \quad (32)$$

For the previous example, where $\xi^{AT} = 10^{-7}$, the limit on the error rate after repeated MMR is $\eta \sim 10^{-11}$. The existence of this limit could also be an interesting subject for future experiments to validate.

## 7. Discussion

When an additional base pair is taken into account, such as A–C, the introduction of a new recognition error, $\zeta^{AC}$, becomes necessary. To account for this additional error, the existing errors $\zeta^{AT}$ and $\zeta^{GT}$ should have to become slightly bigger. Enz needs to strike a balance among these recognition errors when all types of base pairs are involved. Enz should aim to recognize different types of mismatches with similar accuracies [50]. Similarly, it should aim for consistent recognition of correct matches.

Another important factor that influences fidelity is the interactions between neighboring base pairs [51]. The base pair to be recognized is not independent but rather embedded within the DNA through interactions with its neighboring base pairs. These interactions can have an impact on the dynamics of Enz/complex. As a result, the accuracy of recognition may vary slightly depending on the types of neighboring base pairs.

Energy plays a dual role in improving DNA recognition: enhancing recognition accuracy and increasing recognition speed. To illustrate this, let us consider a scenario where we keep $E_B$ and $E_R$ constant (Figure 2), but decrease $E_W$ by $\Delta E$. This change would result in a narrower energy gap $E_W - E_R$, which, in turn, results in a larger recognition error. However, the number of microscopic states $N_W$ associated with configuration $W$ will experience a multiplicative increase of approximately $\exp\left(\frac{\Delta E}{kT}\right)$, leading to a shortened time duration for the $B \to W$ transition by the same factor. Consequently, there exists a trade-off between recognition speed and accuracy as they compete for the limited energy resources.

Translocation also competes for energy. Recognition aims to minimize $E_R$ (Figures 1 and 2) to maximize recognition accuracy, while translocation seeks to maximize $E_R$ for the fastest translocation speed possible. This conflict can be resolved by employing two specialized enzymes (Enz) working in tandem. The first enzyme is solely responsible for mismatch recognition, while the second enzyme handles directional motion and carries the first enzyme. This arrangement allows the first enzyme to position $E_R$ at its lowest value, ensuring the highest recognition accuracy at the expense of limited mobility beyond Brownian motion. It relies on the second enzyme to provide transport. Meanwhile, the second enzyme adjusts $E_R$ to the appropriate level for achieving the desired translocation speed, synchronizing with the recognition speed of the first enzyme. The collaboration between the two enzymes achieves both the highest recognition accuracy and the highest translocation speed.

The interaction between the complex and the surrounding water can be more complex than initially presumed. However, the calculations, particularly Equations (4) and (7), still hold. To further demonstrate this, let us consider a thought experiment, specifically focusing on the quantum transition from state $B$ to state $R$. First, let us imagine the complex is isolated from the surrounding water but shares the same temperature, for instance, $T = 300$ K (or 27 °C). During the transition, an amount of energy $\Delta E = E_B - E_R$ is released, leading to an increase in the complex's temperature from $T = 300$ K to, for instance, $T' = 310$ K. As a result, the entropy of the complex increases by approximately $\Delta E / T' \approx \Delta E / T$, which aligns with Equations (3) and (4). Next, let us consider the isolated complex covered by a single layer of water molecules. Since the interaction between the water layer and the complex is uncertain, let us explore three scenarios: (I) If the water molecules do not participate in the quantum transition, Equation (4) still applies. Interestingly, the complex and the water layer will have different temperatures immediately after the transition, for instance, 310 K and 300 K, respectively. Heat will subsequently flow from the complex to the water layer, resulting in a common temperature, for instance, 308 K. (II) If the water layer fully participates in the quantum transition, it becomes an integral part of the complex. As a result, the complex has more atoms, causing a lesser increase in temperature. It would reach $T' = 308$ K. Hence, we can still use $\Delta E / T' \approx \Delta E / T$ and derive Equation (4). Interestingly, no heat flows between the complex and the water layer afterward, as they are already at the same temperature. (III) If the water layer partially participates in the quantum transition, an intermediate scenario arises in which Equation (4) still holds, just as it does in all extreme scenarios. In this intermediate scenario, the quantum

transition still results in different temperatures for the complex and the water layer, for instance, 309 K and 304 K, respectively. Consequently, heat flows from the complex to the water layer, resulting in a common temperature of 308 K. Finally, let us consider the scenario where multiple layers of water molecules cover the complex, as if the complex is immersed back into the water. In this case, the same analysis applies, and therefore, Equation (4) is applicable.

A fascinating comparison can be drawn between the recognition of a mismatch in DNA and the biological detection of the Earth's magnetic field [52]. Both processes involve information processing, but they are unlikely to share the same underlying mechanisms. In the case of magnetosensitivity, the favored mechanism is the radical-pair mechanism [53]. This mechanism relies on the influence of the Earth's magnetic field on the spin state of a radical pair. Multiple radical pairs can participate in sensing the Earth's magnetic field, leading to more precise sensing results. Fluctuations in individual radical pairs can be averaged out, enhancing the overall sensitivity. In contrast, the recognition of a base pair in DNA is spatially constrained. Only molecules in direct contact with the base pair can participate in the recognition process. In our study, we focus on understanding the functional constraints of Enz without specifying its structure, which is assumed to have been optimized through evolution in a normal environment, meaning that Enz's functions could be influenced by extreme environments, such as a strong magnetic field [54,55].

## 8. Artificial Enz

Enz might be artificially constructed for various purposes. Imagine an artificially constructed Enz that recognizes two types of molecules, denoted as $\alpha$ and $\beta$, respectively, by adopting different configurations. Let us simplify the characteristics of Enz as depicted in Figure 6 so that we can summarize the essence of the proposed mechanism for molecular recognition. In this scenario, we will assume the following conditions: (1) Gravity and water are absent, and all molecules freely float inside an empty container at room temperature $T$; (2) Enz shares the same temperature as the container before and long after a transition occurs, due to radiation; (3) Enz is energized, for example, by ATP, upon combining with a molecule to be recognized.

$$
\begin{array}{lll}
B \rule[0.5ex]{2cm}{0.4pt} \quad E_B^0 \quad N_B^0(T) & \qquad B \rule[0.5ex]{2cm}{0.4pt} \quad E_B^\alpha \quad N_B^\alpha(T) & \qquad B \rule[0.5ex]{2cm}{0.4pt} \quad E_B^\beta \quad N_B^\beta(T) \\[2ex]
& \qquad W \rule[0.5ex]{2cm}{0.4pt} \quad E_W^\alpha \quad N_W^\alpha(T) & \qquad R \rule[0.5ex]{2cm}{0.4pt} \quad E_R^\beta \quad N_R^\beta(T) \\[2ex]
R \rule[0.5ex]{2cm}{0.4pt} \quad E_R^0 \quad N_R^0(T) & \qquad R \rule[0.5ex]{2cm}{0.4pt} \quad E_R^\alpha \quad N_R^\alpha(T) & \qquad W \rule[0.5ex]{2cm}{0.4pt} \quad E_W^\beta \quad N_W^\beta(T) \\
W \rule[0.5ex]{2cm}{0.4pt} \quad E_W^0 \quad N_W^0(T) & &
\end{array}
$$

$$\text{Enz} \qquad\qquad\qquad \text{Enz} + \alpha \qquad\qquad\qquad \text{Enz} + \beta$$

**Figure 6.** Characteristics of an artificially constructed Enz. Enz here has three distinct configurations: $B$, $R$, and $W$. Upon combining with molecule $\alpha$, configuration $W$ possesses a higher chemical energy than configuration $R$, i.e., $E_W^\alpha > E_R^\alpha$. Conversely, when Enz combines with molecule $\beta$, configuration $R$ has a higher chemical energy than configuration $W$, i.e., $E_W^\beta < E_R^\beta$. It is important to emphasize that Enz is a large molecule composed of thousands of atoms, and these atoms undergo thermodynamic vibrations in various ways, resulting in different microscopic states. $N$ represents the number of microscopic states. For instance, $N_W^\alpha(T)$ represents the number of microscopic states associated with configuration $W$ when combined with molecule $\alpha$ at temperature $T$.

When Enz combines with molecule $\alpha$, there are two potential transitions that can take place: $B^\alpha \to W^\alpha$ and $B^\alpha \to R^\alpha$, as illustrated in Figure 7. Similarly, when Enz combines with molecule $\beta$, two other potential transitions can occur: $B^\beta \to W^\beta$ and $B^\beta \to R^\beta$, as depicted in Figure 8.

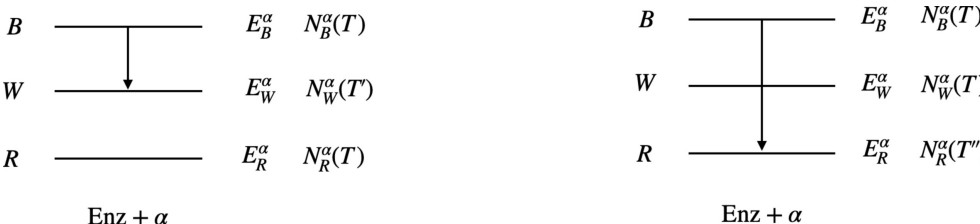

**Figure 7.** Two possible transitions when Enz combines with molecule $\alpha$. In the transition $B^\alpha \to W^\alpha$, a portion of the chemical energy, specifically $E_B^\alpha - E_W^\alpha$, is converted into thermal energy. As a result, the temperature of Enz undergoes a jump from $T$ to $T'$. In the transition $B^\alpha \to R^\alpha$, the chemical energy released is $E_B^\alpha - E_R^\alpha$. In this case, the temperature of Enz jumps from $T$ to $T''$.

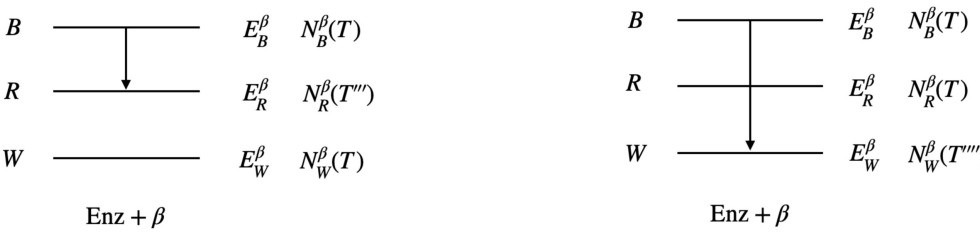

**Figure 8.** Two possible transitions when Enz combines with molecule $\beta$.

If the transition $B^\alpha \to W^\alpha$ occurs, the temperature of Enz jumps from $T$ to $T'$. Similarly, if the transition $B^\alpha \to R^\alpha$ occurs, the temperature jumps from $T$ to $T''$. Although $T'' > T' > T$, these temperatures can be considered approximately equal, i.e., $T'' \approx T' \approx T$.

In the transition $B^\alpha \to W^\alpha$, the chemical energy released is $E_B^\alpha - E_W^\alpha$. The thermal energy of Enz increases by the same amount. As a result, the entropy of Enz increases by approximately $(E_B^\alpha - E_W^\alpha)/T$, and the number of microscopic states of Enz increases by a factor of approximately $\exp\left(\frac{E_B^\alpha - E_W^\alpha}{kT}\right)$. A similar discussion applies to the transition $B^\alpha \to R^\alpha$. Therefore, we can write

$$N_W^\alpha(T') \approx N_W^\alpha(T) \exp\left(\frac{E_B^\alpha - E_W^\alpha}{kT}\right), \tag{33}$$

$$N_R^\alpha(T'') \approx N_R^\alpha(T) \exp\left(\frac{E_B^\alpha - E_R^\alpha}{kT}\right). \tag{34}$$

The probabilities of the transitions $B^\alpha \to W^\alpha$ and $B^\alpha \to R^\alpha$ are given by

$$P(B^\alpha \to W^\alpha) \propto \sum_{j=1}^{N_W^\alpha(T')} |\langle W_j^\alpha | H | B_i^\alpha \rangle|^2, \tag{35}$$

$$P(B^\alpha \to R^\alpha) \propto \sum_{j=1}^{N_R^\alpha(T'')} |\langle R_j^\alpha | H | B_i^\alpha \rangle|^2, \tag{36}$$

where $H$ is the Hamiltonian, $B_i^\alpha$ is the initial microscopic (quantum) state, and $W_j^\alpha$ and $R_j^\alpha$ are the possible final microscopic (quantum) states. In these transitions, the final state has less chemical energy but more thermal energy, ensuring the conservation of total energy.

It is possible that

$$N_R^\alpha(T) \approx N_W^\alpha(T); \quad \langle W_j^\alpha | H | B_i^\alpha \rangle \approx \langle R_k^\alpha | H | B_i^\alpha \rangle, \quad i, j, k = 1, 2, \cdots \tag{37}$$

Therefore, we can write

$$\frac{P(B^\alpha \to W^\alpha)}{P(B^\alpha \to R^\alpha)} \approx \frac{N_W^\alpha(T')}{N_R^\alpha(T'')} \approx \exp\left(-\frac{E_W^\alpha - E_R^\alpha}{kT}\right) \sim 0. \tag{38}$$

This implies that Enz predominantly takes the *R* configuration when combined with molecule *α*. Similarly, for the combination of Enz with molecule *β*, we can write

$$\frac{P(B^\beta \to R^\beta)}{P(B^\beta \to W^\beta)} \approx \frac{N_R^\beta(T''')}{N_W^\beta(T'''')} \approx \exp\left(-\frac{E_R^\beta - E_W^\beta}{kT}\right) \sim 0. \tag{39}$$

Therefore, Enz predominantly takes the *W* configuration when combined with molecule *β*. We can conclude that Enz recognizes molecule *α* and *β* by adopting configurations *R* and *W*, respectively, with only rare mistakes. For instance, when $E_R^\alpha = E_W^\beta = 0$ and $E_W^\alpha = E_R^\beta = 20\,kT$, the recognition errors are approximately $\exp(-20) \sim 10^{-9}$.

Enz can be seen as a specialized version of Maxwell's demon with two interconnected aspects: (1) energy consumption and (2) the potential for making errors. By investing more energy, the likelihood of errors can be reduced, thereby improving accuracy. This relationship applies to Enz composed of various numbers of atoms. However, Equations (38) and (39) are specifically applicable to Enz composed of a large number of atoms.

Our model can be compared to Fermi's golden rule, which is commonly used to study the decay of unstable particles. However, the application of Fermi's golden rule requires detailed knowledge of the Hamiltonian, as well as the specific characteristics of the initial and final states involved. In the case of mismatch recognition, we did not require such detailed information. Instead, we focused on analyzing the relative ratios of Enz transitioning into different conformations. These ratios are typically very small, and our objective was to determine their order of magnitude. This allowed us to identify the dominant factor influencing the ratio, which in our model is the degeneracy obtained from thermodynamics.

Enz's quantum transition can be compared to particle decay; both processes are governed by quantum mechanics. However, they differ in some aspects. In particle decay, an unstable particle transforms into multiple other particles. The initial energy stored in the unstable particle is redistributed as kinetic energy among the final state particles, causing them to move away from the decay point with varying velocities and energies. On the other hand, in a quantum transition process such as $B^\alpha \to W^\alpha$, the final state remains Enz, a cohesive entity formed by numerous atoms bonded together. The released chemical energy, $E_B^\alpha - E_W^\alpha$, cannot manifest as kinetic energy. Instead, it transforms into thermal energy shared among all the atoms.

Chemical energy and thermal energy are separable in Enz. As a physical complex, Enz exhibits numerous macroscopically indistinguishable eigenstates. However, many of these distinct eigenstates arise due to the influence of non-zero temperature. If Enz's temperature were lowered to absolute zero, the number of eigenstates would decrease, possibly resulting in only a few remaining eigenstates corresponding to different configurations and energies. We refer to this energy as chemical energy, which can be determined, in principle, by solving the Schrödinger equation. When Enz's temperature is restored to room temperature, energy is required, which becomes the thermal energy of Enz. Thermal energy contributes to entropy and degeneracy, leading to an increased number of eigenstates. Fortunately, the thermal energy and chemical energy can be studied separately, particularly when they are at different orders of magnitude, thereby minimizing interference. The thermal energy associated with each degree of freedom is typically on the order of $kT$. In contrast, the chemical energy is on the order of $10\,kT$ at room temperature. Consequently, the level of chemical energy is rarely excited by thermal energy, ensuring that the corresponding configuration is minimally affected by thermal fluctuations. However, if the temperature of Enz becomes too high, the thermal activity can excite the chemical energy level. This can lead to undesired changes in the configuration of Enz or even denaturation.

Our model revolves around a mechanism in which minor variations in a molecule to be recognized can result in a significant change in Enz's chemical energy levels. Let us refer to this mechanism as a "quantum lever". Simulations can be conducted to investigate this mechanism, specifically at absolute zero temperature, where thermal effects do not exist .

Once we discover a "quantum lever", it might be constructed in experimental settings and it should function even at temperatures above absolute zero.

Our model is based on the principles of quantum mechanics and thermodynamics. Given the scale at which Enz operates, classical mechanics does not apply. If Enz were to rely on classical mechanics, it would require a level of complexity and sophistication comparable to a macroscopic detector. From a thermodynamic perspective, there are similarities between Enz and macroscopic detectors, such as computer reading heads, in terms of their functionality. Both Enz and macroscopic detectors consume energy and are susceptible to errors. Therefore, studying Enz can provide insights into the energy efficiency limitations of macroscopic detectors when it comes to achieving a specific level of detection accuracy.

### 9. Conclusions

Firstly, we proposed that MMR needs to recognize both correct matches and mismatches, as the overall fidelity of DNA replication relies on the accuracy of both processes. Subsequently, we demonstrated that the contribution of MMR to fidelity can be quantitatively determined using principles from physics. Given the ongoing investigation into MMR enzymes, we focused our study on a hypothetical enzyme called Enz, which actively recognizes base pairs by utilizing ATP. In our proposed model, Enz functions as a quantum system with discrete energy levels that are strongly influenced by the base pair being recognized. Enz leverages this influence to accurately identify the base pair. The recognition process involves a quantum transition. The outcome of this transition determines the recognition result. Increased energy investments result in improved accuracy in recognition. To account for the contribution of MMR to the fidelity of DNA replication, an energy input exceeding 20 $kT$ is required. Only ATP can provide such a high energy input at room temperature, as the free energy associated with equilibrium is insufficient. This suggests that active recognition, rather than passive recognition, is responsible for MMR. Similar mechanisms may also exist in the other two steps of DNA replication, i.e., base selection and proofreading. Furthermore, Enz can be viewed as a specific version of Maxwell's demon. It operates without dedicated memory and instead relies on real-time energy expenditure.

**Author Contributions:** Conceptualization, N.Z. and Y.Z.; methodology, N.Z. and Y.Z.; validation, N.Z. and Y.Z.; formal analysis, N.Z. and Y.Z.; writing—original draft, N.Z. and Y.Z.; writing—review and editing, N.Z. and Y.Z. All authors have read and agreed to the published version of the manuscript.

**Funding:** This research received no external funding.

**Data Availability Statement:** Not applicable.

**Conflicts of Interest:** The authors declare no conflict of interest.

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
