# Peer review of "ATP-Dependent Mismatch Recognition in DNA Replication Mismatch Repair"

_quantumrep, doi:10.3390/quantum5030037_

Round 1
Reviewer 1 Report (New Reviewer)
The paper discusses the role of mismatch repair in DNA replication and the potential involvement of non-equilibrium and quantum processes in improving mismatch recognition accuracy. The authors raise that the mechanism for recognising mismatches needs to be clarified. The core concept of the paper is to introduce a hypothetical quantum enzyme with three energy levels corresponding to incorrect and correct matches. This approach connects energy expenditure and recognition error rate, combining quantum mechanics and thermodynamics.
It is an interesting paper with some out-of-the-box ideas and themes. However, it is relatively speculative, and I could only follow the core concepts of the ideas. There are several avenues the authors can take to improve their article. First and foremost, the physical-chemical mechanism and concept behind quantising the hypothetical enzyme need to be further expanded or discussed. Can you further justify or explain the energy shifts of the mismatch and the canonical pairing leading to repair? Specifically, this statement has an issue when we map back to a real enzyme: (Line 112)"Enz only takes certain configurations, which are eigenstate solutions to the Schrödinger equation; therefore, a change in configuration is a quantum transition." In a physical enzyme, the quantum eigenstates of the system are well-defined at the atomistic level described by quantum chemistry. Still, as you zoom out to a whole enzyme level, you would have loads of eigenstates due to loads of degrees of freedom such that they are no longer macroscopically indistinguishable, resulting in a series of classical conformations. It is not clear to me how this issue is resolved.
The secondary and tertiary structure of the DNA+enzyme complex and the interactions with the enzyme and the DNA is important, as demonstrated experimentally [1] and in simulations for the wobble misincorporation mechanism [2, 3]. Also look at the additional references within [1, 2]. Could the proton transfer variant or the interactions be important or relevant here?
Assuming three enzyme conformations and using a classical mechanism, how does it fail to reproduce the mismatch sensing using your theoretical framework?
Could you offer any physical-chemical insights for simulations or experiments to expand this idea of a hypothetical quantum enzyme?
I pre-emptively thank the authors for their time addressing and including some points I raised and wish them well on their manuscript.
[1] Kimsey, I. J., Szymanski, E. S., Zahurancik, W. J., Shakya, A., Xue, Y., Chu, C. C., ... & Al-Hashimi, H. M. (2018). Dynamic basis for dG• dT misincorporation via tautomerisation and ionisation. Nature, 554(7691), 195-201.
[2] Li, P., Rangadurai, A., Al-Hashimi, H. M., & Hammes-Schiffer, S. (2020). Environmental effects on guanine-thymine mispair tautomerisation explored with quantum mechanical/molecular mechanical free energy simulations. Journal of the American Chemical Society, 142(25), 11183-11191.
[3] Slocombe, L., Winokan, M., Al-Khalili, J., & Sacchi, M. (2022). Quantum Tunnelling Effects in the Guanine-Thymine Wobble Misincorporation via Tautomerism. The Journal of Physical Chemistry Letters, 14(1), 9-15.
Some very minor sentence syntax issues.
Author Response
Please see the attachment.

Reviewer 2 Report (New Reviewer)
Due to mistakes in proofreading abilities of polymerases, incorrect bases can be incorporated into our genome leading to mismatches, which in turn can result to mutations. Enzymes from mismatch repair (MMR) family improve the replication fidelity by 1000-fold, thus helping in reducing the errors in DNA replication. Since the knowledge on mismatch repair mechanisms is limited, the authors have used a hypothetical enzyme (Enz) as a quantum system to illustrate the possible MMR mechanism. This quantum system has three energy levels (B, R and W) which involve different conformations of the enzyme. MMR mechanism uses energy from ATP molecules to start recognition at the high energy state (B) and the resulting energy release is used to drive the quantum transition which either leads to a low energy state with correct base pair (R) or a mismatch pair (W). The authors have proposed that the energy gap between these two low layers can be exploited to increase the recognition accuracy and speed of MMR enzymes.
Overall, it is a very detailed study. The results are presented clearly with appropriate reference to figures. This is a valuable study in mismatch repair recognition mechanism area as the research on MMR is limited, this study may inspire others to explore this area further to reveal new aspects of MMR.
The article is suitable for the quantum reports journal.
A few minor points should be addresses before publishing. Further review is not needed.
Line 2: The authors talk about ‘the mismatch recognition mechanism not being established’. The sentence seems a bit vague. Is the recognition mechanism not established for any repair enzyme? If so, adding that detail into the sentence would clarify the point.
Lines 38-39: ‘However, interestingly, different MMR pathways increase DNA replication fidelity by a similar amount’: The sentence is too vague for an audience not versed with MMR field. How can enzymes with different sequences can have same replication efficiency? Adding one more sentence to clarify the current sentence further would help.
Line 1: If ‘greatly’ can be replaced with ‘significantly’, the sentence would flow easier.
Line 27: Replace ‘ATPase sites at the top of the upper channel’ with ‘ATPase sits at the top of the upper channel’.
Author Response
Please see the attachment.

Reviewer 3 Report (New Reviewer)
Overall, the manuscript presents an interesting study on the role of DNA mismatch repair (MMR) in DNA replication fidelity. The authors propose a model based on a hypothetical enzyme, Enz, and utilize principles from physics and quantum mechanics to explain the recognition process and energy requirements for accurate base pair recognition. The manuscript is well-written and provides valuable insights into the physical aspects of MMR. I recommend accepting the manuscript with minor revisions.
Specific comments: The introduction provides a clear background and rational for the study. However, it would be helpful to briefly summarize the main findings or contributions of previous studies on MMR to provide context for the proposed model. The proposed model of Enz as a quantum system and its connection to Maxwell's demon is intriguing. However, it would be beneficial to discuss the limitations or assumptions of this model. Are there any experimental or theoretical studies that support or challenge the proposed model? Including such discussions would enhance the manuscript. The calculations and energy considerations presented in the manuscript are insightful. However, it would be helpful to explain how these calculations were performed in more detail. What parameters were used, and how were they obtained? Providing this information would improve the transparency and reproducibility of the study. The manuscript could benefit from the inclusion of more experimental evidence or data that support the proposed model. While the hypothetical nature of Enz is acknowledged, are there any experimental observations or findings that align with the proposed quantum transition process and energy requirements? Addressing this point would strengthen the manuscript's arguments. The conclusion section provides a concise summary of the study's key findings. However, it would be valuable to discuss potential implications or applications of the proposed model. How might this model influence our understanding of DNA replication fidelity or inspire future research directions? Expanding upon these aspects would enhance the manuscript's significance.
Questions for the Authors:
00001.
1/ How does the proposed model of Enz as a quantum system align with existing experimental findings or observations related to DNA mismatch repair? Are there any specific experiments that support or challenge the quantum transition process and energy requirements proposed in the model?
2/ Considering the hypothetical nature of Enz, are there any plans or possibilities for experimental validation of the proposed model? How might future experiments contribute to our understanding of the physical aspects of DNA mismatch repair and replication fidelity?
3/ In the discussion of the proposed model, you mentioned the limitations or assumptions. Could you elaborate on these limitations and discuss any potential alternative models or mechanisms that could explain the observed phenomena?
4/ Beyond its implications for our understanding of DNA replication fidelity, are there any potential practical applications or future research directions that stem from the proposed model? How might this model inspire new experiments or investigations in the field?
Please address these comments and questions in the revised manuscript. Overall, the study is valuable and provides valuable insights into the physical principles underlying DNA mismatch repair.
Round 2
Reviewer 1 Report (New Reviewer)
You have addressed a good revised version of the manuscript and it can be accepted in the current format.
This manuscript is a resubmission of an earlier submission. The following is a list of the peer review reports and author responses from that submission.
Round 1
Reviewer 1 Report
ATP-dependent mismatch recognition in DNA replication mismatch repair Nian-Qin Zhang and Yong-Jun Zhang
I've thought about this paper several times and come to the conclusion that I think it is not fundamentally sound. The paper investigates a scheme for "active recognition" of errors in DNA replication vs. "passive recognition." The authors believe that "non-equilibrium" active recognition, powered by ATP as a source of free energy, can be superior to "equilibrium" passive recognition, the latter of which they claim is the usual concept employed to understand DNA mismatch recognition, itself not very well understood.
First, let me say that I'm not convinced that equilibrium passive recognition will work at all. The reason is that error recognition involves information, and I don't see how this can avoid free energy expenditure. On p. 2 the authors mention Brownian motion and the long time scale on which it might work. My point is that Brownian motion may only work when it is made direction or "rectified," and my understanding is that this only happens with free energy input, e.g. as in Brownian motors.
So, it may be that their principal focus, nonequilibrium recognition, may be the only thing that will work on a finite timescale. So far so good, it sounds like they have an interesting idea! But my objection to their proposal is in the details of how it would work. The basic idea seems to be in Fig. 2, which shows transitions to R and W quantum levels from an upper level, which is reached with the help of ATP free energy expenditure. (Do R and W stand for Right and Wrong?) The claim is that transitions are more likely to the lower levels, R in the case of Fig. 2a, and W' in the case of Fig. 2b. But I don't see how this is guaranteed at all. There are lots of cases where transition probabilities to higher levels are greater than to lower levels, unlike the assumption with Fig. 2b.
So at a fundamental level, it seems to me their idea, which seemed promising, fails in a very basic way. The authors will have to do some work to convince me that I am wrong. Or, perhaps I am wrong, and just don't see it, and they will be able to convince others.
Reviewer 2 Report
Please see the attached file.

Round 2
Reviewer 1 Report
Review of revision:
The authors have amended Fig. 2 to take into account my serious concerns about the functional properties they attribute to the complex by virtue of its energy levels. They now have substituted numbers of energy levels associated with degeneracies. That might be OK, except, read on.
Then, they make a key argument about degeneracy (as mentioned above) of the complex in various states B, R, W, and B', R', W'.
I still have a lot of problems with their scheme for error detection.
It is very unclear to me what they are doing with 2 units of ATP in Fig. 1. Using the energy from the first unit raises the energy of the "Enz" complex from S to B. This makes sense if the ATP is held to release energy, as is usually the case. But then the second ATP unit is held to release energy in the transition from B. This makes no sense to me at all!
The authors also claim that the configurations of the complex are "quantum eigenstates." But how can this be? The complex is in water – therefore its eigenstates cannot be considered separately from the water medium. Do they mean an eigenstate of the complex considered in isolation? Maybe OK. But see the remarks below about "doorway" states.
They also talk about energy being released apparently from the complex, then is absorbed by the complex. This makes no sense to me. Then they have a very confusing claim about the temperature of the complex increasing. It seems to me that if the complex releases heat in going from one complex energy level to a lower one, that heat will simply be released to the surrounding water – all at a common temperature, at least after all the relevant processes are complete.
They also make arguments about probabilities based on state counting. This is based on a kind of quasi-ergodic hypothesis in which all transitions involving degenerate states are equally probable. This assumption is used to compute probabilities e.g. Eqs. 6, 8. This is fine for thermodynamics, but what about rates? The transition to the W "doorway" state might be very fast compared to that to R. If the error correction is supposed to work on this fast time scale, it seems it would lead to error in this scenario!
Also, the state-counting seems very problematic to me. It is based on entropy equations like Eq. 3. Einstein made a similar argument in a lecture once about Brownian motion. The total system (Brownian particle plus water) has more states when the Brownian particle is in a lower gravitational configuration (lower in the water). From this one can easily get the Boltzmann factor. But it absolutely depends on including the water in the state count! The authors' procedure seems to me to be very confused about this.
In summary, I just see a lot of problems with this paper. At best, it needs a lot more work on the basic assumptions. And I have not even gotten to the second part of the paper on shuffling of energy levels.
The authors are trying to come up with a highly speculative mechanism of error correction. That is OK, but the assumptions and mechanism need to be very clearly and cleanly laid out.
I have read the remarks of "Referee 2" and I am OK with overlooking them – though Referee 2 might not be! It seems to me the authors are trying to come up with a highly speculative mechanism of error correction. The qualms of Referee 2 about magnetic systems, spin etc. seem to me not very relevant.
Reviewer 2 Report
I am glad that the authors found my comments insightful and tried to implement them into their work. However, this is an issue with the last line in the last paragraph of the Discussion section, "A single molecule does not have to have a high magnetic sensitivity, which the collective effect of a large number of molecules can achieve. In contrast, recognizing a mismatch can only be performed by a single enzyme due to the base pair's small size; therefore, the enzyme has to be very sensitive to the base pair." To be clear, in most biological environments, there are numbers of the same molecules. So a statement such that is wrong at best is confusing. Furthermore, a new study just published in Nature (Essential elements of radical pair magnetosensitivity in Drosophila https://doi.org/10.1038/s41586-023-05735-z) shows that the FAD molecule is essential for magnetosensing in Drosophila. This issue needs to be addressed appropriately.
The reference numbers are not available anymore, instead showing up by '?' sin.
Round 3
Reviewer 1 Report
Basically, I don't think the paper should be published. If it is my decision, that is my verdict! It pains me that the authors are going to all this work without being able to persuade me that they know what they are doing. Their responses just don't convince me.
Reviewer 2 Report
The authors have addressed my concerns. I have no further comments.
